# Electric-dipole transitions in $^6$Li with a fully microscopic six-body calculation

**Wataru Horiuchi$^\star$ and Shuji Satsuka**

Department of Physics, Hokkaido University, Sapporo 060-0810, Japan

$\star$ whoriuchi@nucl.sci.hokudai.ac.jp

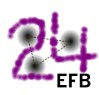 *Proceedings for the 24th edition of European Few Body Conference, Surrey, UK, 2-6 September 2019*

## Abstract

**Exploring new excitation modes and the role of the nuclear clustering has been of great interest. An interesting speculation was made in the recent photoabsorption measurement of $^6$Li that implied the importance of the nuclear clustering. To understand the excitation mechanism of $^6$Li, we perform a fully microscopic six-body calculation on the electric-dipole ($E1$) transitions and discuss how $^6$Li is excited by the $E1$ field as a function of the excitation energy. We show the various cluster components in the six-body wave functions and discuss the role of the nuclear clustering in the $E1$ excitations of $^6$Li.**

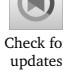
## 1 Introduction

Nuclear cluster structure often appears in the spectrum of light nuclei. The role of the nuclear clustering in the electric-dipole ($E1$) transitions has been of great interest since they are closely related to important astrophysical reactions. Recently, an interesting speculation on the $E1$ excitation mechanism was made in the measurement of the photoabsorption cross section of $^6$Li that implied the coexistence of the typical and cluster $E1$ excitation modes [1]. To understand this excitation mechanism, we performed a fully microscopic six-body calculation with the correlated Gaussian method, in which the formation and distortion of the nuclear clusters are naturally taken into account as was demonstrated for $^6$He [2]. Basically, this contribution aims at reviewing the discussions and findings given in Ref. [3] accompanying with some unpublished results. We calculate the $E1$ transition strengths and their transition densities and discuss how $^6$Li is excited by the $E1$ field as a function of the excitation energy. We find the out-of-phase transitions due to the valence proton and neutron around the alpha cluster dominate in the low-lying energy regions below the alpha breaking threshold indicating "soft" Goldhaber-Teller (GT [4]) excitation, which is very unique in $^6$Li, whereas the typical GT mode appears in the higher-lying energy regions. We discuss the role of the nuclear clustering

in the $E1$ excitations of $^6$Li by showing the various cluster components in the six-body wave function.

In Sec. 2, we describe the setup of the microscopic six-body calculation. How we construct the six-nucleon wave functions are briefly explained. Section 3 presents calculated results and discusses the structure of the $E1$ excitations of $^6$Li through the $E1$ transition strengths in Sec. 3.1 and its transition densities in Sec. 3.2. In Sec. 3.3, the isoscalar dipole transitions are discussed as complementary information of the ordinary $E1$ excitation. Conclusion is made in Sec. 4.

# 2 Microscopic six-body calculation for $^6$Li

## 2.1 Hamiltonian

The Hamiltonian for a six-nucleon system is set by the kinetic energy and two-body nuclear and Coulomb potential terms in the center-of-mass (cm) frame. For the sake of simplicity, we employ the Minnesota (MN) nucleon-nucleon potential [5] which offers a fair description of the binding energies and radii of $s$-shell nuclei, $^2$H ($d$), $^3$H ($t$), $^3$He ($h$), and $^4$He ($\alpha$) [6, 7] without a three-body force. The MN potential includes the one parameter $u$ responsible for the strength of odd-parity partial waves. Roughly speaking, the $u$ parameter controls the interaction of the valence nucleons from the $\alpha$ core on the $p$-shell orbital. Since the original strength ($u = 1.00$) does not give the correct low-lying threshold and rms radius of $^6$Li, the other two parameter sets are examined that are chosen to reproduce the three-body threshold of $\alpha + p + n$ ($u = 0.93$) and the rms radius of $^6$Li ($u = 0.87$), respectively.

## 2.2 Correlated Gaussian expansion

To describe the six-nucleon ground and excited states, we expand the spatial part of the wave function in terms of the correlated Gaussian function with the global vector representation [7, 8]: $\exp\left(-\sum_{j,k=1}^{5} A_{jk} \boldsymbol{x}_j \cdot \boldsymbol{x}_k\right) \mathcal{Y}_{LM_L}(\sum_{j=1}^{5} v_j \boldsymbol{x}_j)$. The coordinate set $(\boldsymbol{x}_1, \ldots, \boldsymbol{x}_5)$ excluding the cm coordinate of the six-nucleon system, $\boldsymbol{x}_6$, is taken as the Jacobi coordinate. The correlations among the particles are explicitly included through the off-diagonal parameter, $A_{jk}$ ($j \neq k$). The rotational motion of the system is described with the so-called global vector in the argument of the solid spherical harmonic [8, 9]. This expression is convenient that the functional form does not change under any linear transformation of the coordinate. Various configurations such as single-particle, $\alpha + p + n$ and $h + t$ cluster configurations can easily be implemented. The matrix elements of the Hamiltonian can be evaluated analytically. See [6–8] for the detailed mathematical derivation and expressions. With these nice property, the correlated Gaussian has been applied to studying the nuclear clustering [10–12]. See, also review papers [13, 14]. The six-nucleon spin and isospin functions as well as antisymmetrization of the total basis function are fully taken into consideration. The isospin mixing due to the Coulomb interaction is naturally described in this study.

### 2.2.1 Ground-state wave function

To find an optimal choice of a huge number of the variational parameters, we employ the stochastic variational method (SVM) [7, 8]. We increase the number of basis selected competitively from randomly generated candidates until a certain number of basis states are obtained. Figure 1 displays the obtained energy curve with $u = 1.00$ as a function of the number of bases. We see the energy of the six-nucleon system is rapidly converged with increasing the number of the basis functions. We stop the calculation with 600 bases, which are very small

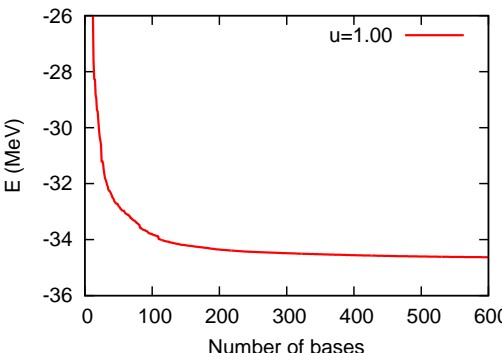

Figure 1: Energy convergence of $^6$Li as a function of the number of bases.

by noting that each basis function includes $6(6-1)/2 = 15$ parameters as well as the spin degrees of freedom. Then we switch the selection procedure for the refinement of the variational parameters in the already obtained basis functions until the energy is converged within tens of keV. For the wave functions with other $u$ parameters, we start with the optimal basis functions with $u = 1.00$ and refine those basis functions by keeping the total number of basis unchanged until the energy convergence is reached.

### 2.2.2 Construction of excited states

Here we overview how to construct the final-state wave functions excited by the $E1$ operator: $\mathcal{M}(E1, \mu) = e\sqrt{\frac{4\pi}{3}} \sum_{i \in p} \mathcal{Y}_{1\mu}(\boldsymbol{r}_i - \boldsymbol{x}_6)$, where $\boldsymbol{r}_i$ is the single-particle coordinate of the $i$th proton. All the details are described in Ref. [3]. We expand the final-state wave function in a large number of the correlated Gaussian basis functions. To incorporate the six-body correlations efficiently, physically important configurations are selected: (I) Single-particle excitation, (II) 4+1+1 cluster, and (III) 3+3 cluster configurations.

The configurations of type (I) is based on the idea that the $E1$ operator excites one coordinate in the ground-state wave function. The resulting coherent states are important to satisfy the $E1$ sumrule [2, 15, 16]. They are constructed by using the basis set of the ground-state wave function of $^6$Li by multiplying an additional solid spherical harmonic with the angular momentum 1. The configurations of types (II) and (III) explicitly describe the cluster configurations of $\alpha + p + n$ and $h + t$, which correspond to the two lowest thresholds. The relative wave functions of the valence nucleons are expanded with several Gaussian functions covering from short to far distances up to about 20 fm.

We include all the basis states for each subsystem independently. Therefore, the final-state wave functions are not restricted to the subsystems being the ground state but the excitations and distortion of $^6$Li, $\alpha$, $h$, and $t$ are included through the coupling of the pseudo excited states of those nuclear systems. We diagonalize the Hamiltonian including all the configurations of types (I)–(III) with 18490 basis functions and find $\sim 2 \times 10^3$ states below the excitation energy of 100 MeV.

Though we employed the large model space mentioned above, the states are discretized since the wave functions are expanded by square-integrable bases. For more quantitative discussions, it is necessary to include the continuum effect explicitly by employing, e.g., the complex scaling method [17] and Lorentz integral transform method [18]. However, this is involved and the beyond the scope of the present analysis.

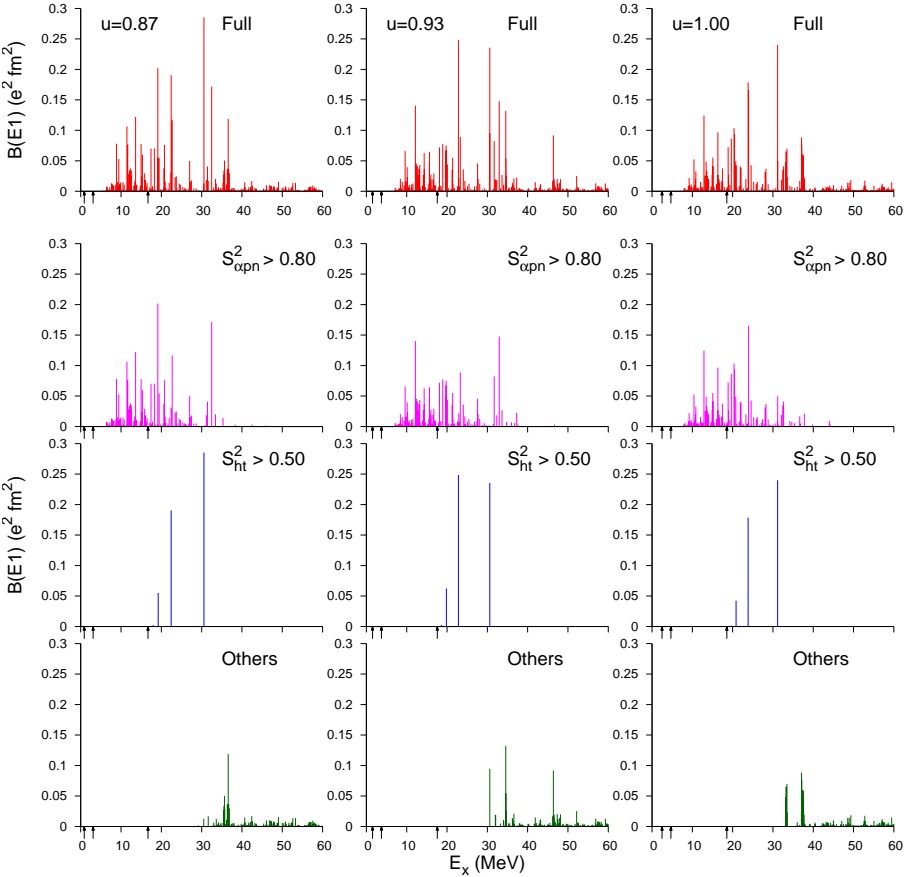

Figure 2: Electric-dipole transition strengths and their decomposition with respect to the spectroscopic factors. The parameters $u = 0.87, 0.93$ and $1.00$ in the MN interaction are employed. Arrows indicate the theoretical threshold energies of $\alpha+d$, $\alpha + p + n$, and $h + t$ from left to right, respectively. The results with $u = 0.93$ and the Full calculations are adopted from Ref. [3].

## 3 Results and discussions

### 3.1 Electric-dipole transition strengths

Figure 2 plots the $E1$ transition strengths or reduced transition probability $[B(E1)]$ obtained with the full model space that includes the configurations of types (I)–(III) with different values of the $u$ parameter as a function of the excitation energy, $E_x$. Large $B(E1)$ values in the low($E_x \lesssim 16$ MeV)-, intermediate($E_x \sim 16$–30 MeV)-, and high($E_x \gtrsim 30$ MeV)-energy regions are found. There is little quantitative difference among these three different $u$ values up to $E_x \sim 40$ MeV.

The decomposition of the $E1$ transition strengths with respect to the spectroscopic factors of $\alpha + p + n$ ($S_{\alpha p n}^2$) and $h + t$ ($S_{ht}^2$) are also plotted in Fig. 2. Here $S_{\alpha p n}^2$ and $S_{ab}^2$ are respectively defined by $\left|\left\langle \Psi^{(\alpha)}\Psi^{(p)}\Psi^{(n)} \middle| \Psi^{(6)} \right\rangle\right|^2$ and $\left|\left\langle \Psi^{(a)}\Psi^{(b)} \middle| \Psi^{(6)} \right\rangle\right|^2$ with the wave function $\Psi^{(X)}$ of $X(= p, n, d, h, t, \alpha)$ cluster and $X = 6$ denoting the six-nucleon system. In the ground state of $^6$Li, the $S_{\alpha d}^2$ value is large $\sim 0.9$ [3] for all the $u$ parameters employed, indicating a well-developed $\alpha + d$ cluster structure. As we see in the $E1$ transitions to the states with $S_{\alpha p n}^2 > 0.80$, most of the low-lying states below 20 MeV have a large $\alpha + p + n$ cluster compo-

Table 1: Excitation energy $E_x$ in MeV, $B(E1)$ in $e^2\text{fm}^2$ and $\alpha + p + n$ and $h + t$ spectroscopic factors in $^6$Li with different $u$ parameters. The results with $u = 0.93$ are adopted from Ref. [3].

| $E_x$ | $u = 0.87$ $B(E1)$ | $S^2_{\alpha pn}$ | $S^2_{ht}$ | $E_x$ | $u = 0.93$ $B(E1)$ | $S^2_{\alpha pn}$ | $S^2_{ht}$ | $E_x$ | $u = 1.00$ $B(E1)$ | $S^2_{\alpha pn}$ | $S^2_{ht}$ |
|------|------|------|------|------|------|------|------|------|------|------|------|
| 8.9 | 0.076 | 1.000 | 0.000 | 9.6 | 0.066 | 0.999 | 0.000 | 10.4 | 0.052 | 0.999 | 0.001 |
| 11.5 | 0.106 | 0.999 | 0.006 | 12.1 | 0.140 | 0.999 | 0.011 | 12.9 | 0.124 | 0.999 | 0.009 |
| 11.7 | 0.076 | 0.999 | 0.007 | 14.3 | 0.063 | 0.997 | 0.000 | 15.1 | 0.055 | 0.997 | 0.000 |
| 13.6 | 0.122 | 0.997 | 0.000 | 15.7 | 0.064 | 0.999 | 0.010 | 16.3 | 0.097 | 0.998 | 0.016 |
| | | | | | | | | | | | |
| 19.1 | 0.202 | 0.988 | 0.005 | 18.9 | 0.077 | 0.991 | 0.004 | 20.3 | 0.103 | 0.993 | 0.001 |
| 20.8 | 0.076 | 0.994 | 0.010 | 19.8 | 0.075 | 0.994 | 0.003 | 20.5 | 0.094 | 0.995 | 0.001 |
| 22.4 | 0.191 | 0.124 | 0.845 | 22.8 | 0.249 | 0.113 | 0.850 | 23.8 | 0.179 | 0.133 | 0.827 |
| 22.7 | 0.117 | 0.948 | 0.013 | 23.3 | 0.089 | 0.963 | 0.003 | 23.9 | 0.165 | 0.880 | 0.085 |
| | | | | | | | | | | | |
| 30.6 | 0.285 | 0.035 | 0.691 | 30.6 | 0.236 | 0.264 | 0.533 | 31.2 | 0.240 | 0.162 | 0.618 |
| 32.5 | 0.172 | 0.918 | 0.005 | 30.6 | 0.095 | 0.780 | 0.158 | 33.5 | 0.070 | 0.712 | 0.002 |
| 35.7 | 0.050 | 0.594 | 0.002 | 33.0 | 0.148 | 0.962 | 0.005 | 37.1 | 0.088 | 0.045 | 0.001 |
| 36.6 | 0.119 | 0.351 | 0.002 | 34.6 | 0.132 | 0.195 | 0.019 | 37.2 | 0.077 | 0.052 | 0.003 |

nent. The $E1$ strengths with $S^2_{ht} > 0.50$ show three large $E1$ strengths after the $h + t$ threshold 15.8 MeV [19], opens. These robust structures do not depend on the $u$ parameter. The first two can be the observed levels at $E_x = 17.98$ and 26.59 MeV, having relatively small $h + t$ decay widths, 3.0 and 8.7 MeV for the first and second peaks, respectively [19]. The other complementary strengths are plotted in the lowest three panels of Fig. 2. Since all the particle thresholds are open beyond 30 MeV, many small strengths are distributed due to the coupling of various configurations

For quantitative discussions, Table 1 summarizes $E_x$, $B(E1)$, $S^2_{\alpha pn}$, and $S^2_{ht}$ of the states that give four largest $B(E1)$ values in the low-, intermediate-, and high-energy regions with $u = 0.87, 0.93$, and 1.00. We note that in the low energy regions ($E_x \lesssim 16$ MeV) below the $h + t$ threshold, all the states have large $S^2_{\alpha pn}$ values being $\sim 1$, whereas their $S^2_{ht}$ values are $\sim 0$.

## 3.2 Dipole transition densities

To discuss the structure of the $E1$ excitations, here we calculate the transition densities: $\rho^{\text{tr}}_{p/n}(r) = \sum_{i \in p/n} \langle J_f \| \mathcal{Y}_1(\boldsymbol{r}_i - \boldsymbol{x}_6)\delta(|\boldsymbol{r}_i - \boldsymbol{x}_6| - r)\| J_0 \rangle$, where $J_0$ and $J_f$ are the angular momentum of the ground and final states, respectively. Note that the $E1$ transition matrix can be obtained with $\langle J_f \| \mathcal{M}(E1) \| J_0 \rangle = e\sqrt{\frac{4\pi}{3}} \int_0^\infty dr\, \rho^{\text{tr}}_p(r)$, which express the spatial distribution of the $E1$ transitions. In the following, we discuss the transition densities for $^6$Li of the selected transitions that show some characteristic behaviors.

### 3.2.1 Soft Goldhaber-Teller excitations

Figure 3 displays $\rho^{\text{tr}}_p$ and $\rho^{\text{tr}}_n$ to the states that have the prominent $B(E1)$ values. We discuss the results with $u = 0.87$, which were not presented in Ref. [3], though they are qualitatively the same as those obtained with $u = 0.93$. The transition densities such at (a) $E_x = 13.6$, (c) 19.1, and (e) 32.5 MeV can be categorized into the same group. All the transition densities show the in-phase transition around the $^4$He radius and the out-of-phase transitions beyond the nuclear surface. More oscillations in the outside region appear with increasing the excitation energy. We interpreted this unique transition as "soft" Goldhaber-Teller(GT)-dipole mode, that is, the

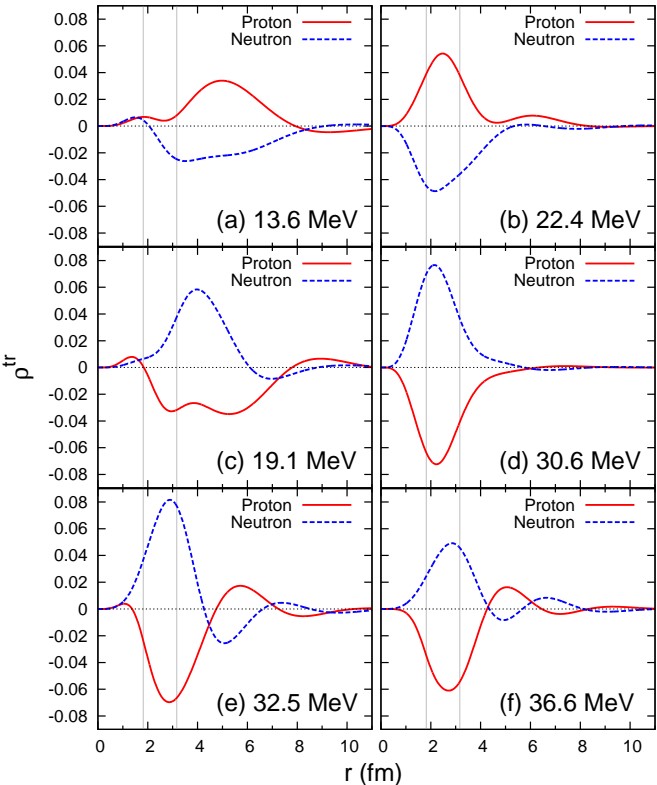

Figure 3: Transition densities for proton and neutron with $u = 0.87$ at (a) 13.6, (b) 22.4, (c) 19.1, (d) 30.6, (e) 32.5, and (f) 36.6 MeV. Vertical thin lines indicate theoretical nuclear radii, $^4$He and $^6$Li, respectively.

oscillation of the valence proton and neutron around the core ($\alpha$) [3], which is a variant of the classical picture of the giant dipole resonance (GDR) [4]. In Table 1, we see all these states a large $S^2_{\alpha pn}$ value $\gtrsim 0.9$. It should be noted that this mode is different from the so-called soft-dipole mode as Ref. [2] showing the oscillation of the valence two neutrons against the core [2].

### 3.2.2 Cluster and Goldhaber-Teller excitations

At (b) $E_x = 22.4$, (d) 30.6, and (f) 36.6 MeV, all the transition densities exhibit out-of-phase transitions in all regions, showing the typical GT mode.

Let us first discuss the states at (b) $E_x = 22.4$ and (d) 30.6 MeV. In this energy region, the $h + t$ threshold already opens. Considering the fact that the $S^2_{ht}$ values are large $\gtrsim 0.7$ (see Table 1), this behavior can be interpreted as the $E1$ excitation of the relative wave function between the $h$ and $t$ clusters. Actually, the peak positions are located at $\sim 2$ fm, which is about the sum of the peak positions of the densities of $h$ and $t$ [3].

We also see the state with the state with (f) $E_x = 36.6$ MeV also show the out-of-phase transition in all regions but the $E1$ excitation structure is found being different from those of (b) and (d). The peak positions are located somewhat outside from these of (b) and (d), and the $S^2_{\alpha pn}$ and $S^2_{ht}$ are small as listed in Table 1. This is nothing but the typical GT mode, in which the protons and neutrons oscillate against each other [4].

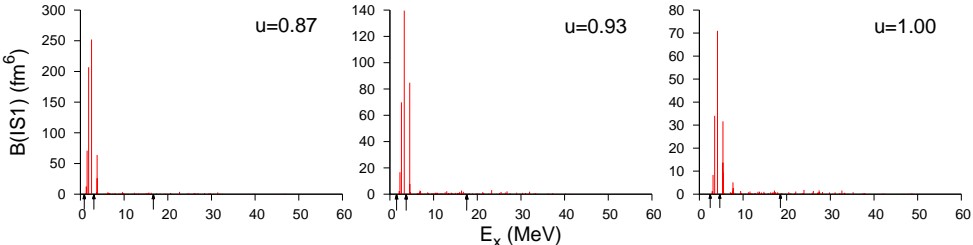

Figure 4: Isoscalar dipole transition strengths with $u = 0.87$, 0.93, and 1.00. The results with $u = 0.93$ are adopted from the data given in Ref. [3].

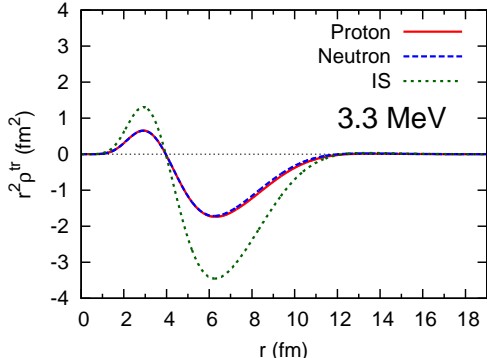

Figure 5: Transition densities for proton and neutron and their sum $\rho_p^{tr} + \rho_n^{tr}$ (IS) multiplied by $r^2$ with $u = 0.93$ at $E_x = 3.3$ MeV.

### 3.3 Discussion: Isoscalar dipole transitions

To discuss more details on the transition densities, here we discuss the isoscalar dipole excitation whose operator is defined by $\mathcal{M}(\text{IS1}, \mu) = \sum_i (r_i - x_6)^2 \mathcal{Y}_{1\mu}(r_i - x_6)$ [20]. By definition, the IS1 transition matrix can be obtained with $\int_0^\infty dr\, r^2 \left( \rho_p^{tr} + \rho_n^{tr} \right)$.

Figure 4 displays the IS1 transition strengths with different values of the $u$ parameter as a function of the excitation energies. Some prominent strengths appear between the $\alpha + d$ and $\alpha + p + n$ thresholds and the $S_{\alpha d}^2$ values of those states are $\sim 1$. They come from the transition to the $\alpha + d$ continuum states, which cannot be excited by the ordinary isovector $E1$ operator. Note that the isospin mixing components due to the Coulomb interaction are included in the wave functions but they are small. Figure 5 draws the transition densities multiplied by $r^2$ that show the most prominent $B(IS1)$ value at $E_x = 3.3$ MeV with $u = 0.93$. The transition densities of proton and neutron coincide each other, showing the in-phase transition in all regions.

Let us see Fig. 3 from a different view point. Figure 6 plots the transition densities of proton and neutron multiplied by $r^2$ and their sum of the states given in Fig. 3. We remind that the IS1 transition matrix is obtained by the sum of the proton and neutron transition densities. When the transition densities exhibit the out-of-phase transitions, they are canceled, resulting in a small IS1 transition matrix. In general, the in-phase transition enhances the IS1 transition matrix. In the case of the soft GT-dipole mode, since the in-phase transition occurs in the short distances, the contributions from these regions become small due to the additional $r^2$ factor through the IS1 operator. Since almost all the IS1 strengths are exhausted in the low-energy regions below $\sim 5$ MeV, only small strengths are found in the higher energy regions.

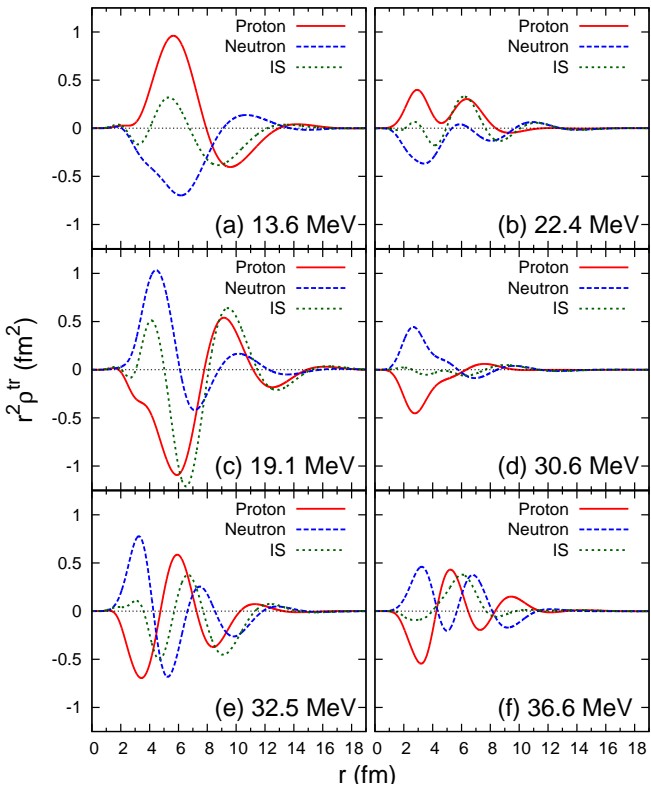

Figure 6: Same as Fig. 5 but with $u = 0.87$ at the same $E_x$ of Fig. 3.

To understand the excitation mechanism of $^6$Li, one can observe this strong suppression of the IS1 transition strengths using such as $(\alpha, \alpha')$ measurement as a complementary evidence of the existence of the soft-GT-dipole mode.

## 4 Conclusion

We performed fully microscopic six-body calculations to understand the electric-dipole ($E1$) excitation mechanism. The six-body wave functions were constructed in terms of the correlated Gaussian (CG) functions. The ground state wave function was obtained precisely with the aid of the stochastic variational method. The final state wave functions were also expressed by a number of the CG functions. The asymptotic wave functions between clusters as well as the distortion of the clusters were taken into account.

We calculate the $E1$ transition strengths and their transition densities. With the analysis of the cluster components of the wave functions, we see that the nuclear clustering emerges in the $E1$ excitation depending on the positions of the threshold energies. We find very unique excitation mode, "soft" Goldhaber-Teller(GT) excitation, which is recognized as the out-of-phase oscillation between valence nucleons around the $\alpha$ cluster in $^6$Li, dominates below the $\alpha$ breaking threshold energy. Series of the vibrational excitation mode are also found in the high-lying energy regions.

Beyond the $h + t$ threshold, the $E1$ transitions with the $h + t$ cluster mode appear. The transition densities show the out-of-phase transition in all regions. With increasing the excitation energy after all decay channels open beyond $\sim 30$ MeV, the typical GT mode appears with largely distorted configurations neither $\alpha + p + n$ nor $h + t$.

Isoscalar dipole (IS1) excitations are examined as it gives a different view of the $E1$ excitation mechanism. The out-of-phase transitions in the soft GT dipole mode cause the strong suppression of the IS1 transition strengths. A measurement of the $\alpha$ inelastic scattering cross sections at around $\alpha + p + n$ may reveal the absence of the isoscalar component at this energy regions, which will be a complementary evidence of the existence of the soft GT dipole mode. Also, exploring the appearance of the soft GT mode in heavier nuclei is interesting, e.g., $^{18}$F as $^{16}$O $+ p + n$.

## Acknowledgements

W. H. acknowledges the collaborative research program 2019, information initiative center, Hokkaido University.

**Funding information** This work was in part supported by JSPS KAKENHI Grants No. 18K03635, No. 18H04569, and No. 19H05140.

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
