# Peer review of "Electric-dipole transitions in 6Li with a fully microscopic six-body calculation"

_SciPost Physics Proceedings, doi:SciPost Phys. Proc. 3, 022 (2020)_

## Round 1 · Referee Report · Anonymous (Referee 1) · 2019-12-11

Report

The paper presents 6-body microscopic calculations of electric dipole transitions in 6Li. It is written in a clear way and gives an insight into the structure of excited 6Li states and its influence on various transition modes. I recommend to publish it after a few clarifications.

Requested changes

1) As I understand the basis used involves wave functions that asymptotically decrease. Therefore, the continuum nature of the excited states is not taken into account. This is why excited states have well-defined energies and B(E1) probabilities are given by lines. This feature of the theory is not mentioned in the text and it would be helpful to make it clear to the reader. Also, it would be nice to comment on the possibility of including the continuum explicitly. 2) What is the definition of the spectroscopic factors used? Is it the same as the widely-accepted one? The definition of the spectroscopic factor should be given. It would also be useful to tell the reader what the spectroscopic factors for the 6Li ground state are. The spectroscopic factors are denoted as S rather than S^2 by those who investigate them experimentally. 3) It would be useful to add experimental 6Li energy in Fig. 1. 4) In first paragraph of Conclusion “asympototic “ - - > “asymptotic”.

  • validity: good
  • significance: good
  • originality: good
  • clarity: good
  • formatting: good
  • grammar: reasonable

Author:  Wataru Horiuchi  on 2019-12-12  [id 677]

(in reply to Report 1 on 2019-12-11)

Dear Referee,

Thank you for your comments. We have revised the text following the most of your suggestions.

1) We have added a paragraph to the end of Sec. 2.2.2, to explain the feature of the theory and the possibility of taking care of the continuum effect.

2) We have written the definition of the spectroscopic factors used in this paper. The values of the spectroscopic factors of the ground state of 6Li have also been added.

3) With $u=1.0$ the ground-state energy of 6Li is not reproduced. To avoid the confusion, we mentioned the ground state energy of 6Li in Sec. 2.1.

4) We have corrected the typo.

Thank you again for taking your time to review our manuscript.

Sincerely, Wataru Horiuchi

---

## Round 4 · Author Response

Dear Referee,

Thank you for your comments. We have revised the text following the most of your suggestions.

1) We have added a paragraph to the end of Sec. 2.2.2, to explain the feature of the theory and the possibility of taking care of the continuum effect.

2) We have written the definition of the spectroscopic factors used in this paper. The values of the spectroscopic factors of the ground state of 6Li have also been added.

3) With $u=1.0$ the ground-state energy of 6Li is not reproduced. To avoid the confusion, we mentioned the ground state energy of 6Li in Sec. 2.1.

4) We have corrected the typo.

Thank you again for taking your time to review our manuscript.

Sincerely, Wataru Horiuchi

---

## Round 4 · List of Changes

This is given in the resubmission letter.

---

## Editorial Decision

published